# Ameliorative effects of *Penthorum chinense* Pursh on insulin resistance and oxidative stress in diabetic obesity *db/db* mice

Jilei Hu[1,2], Leyu Zheng[2,3], Xi Fan[2], Hongmei Lang[4], Huibo Xie[2], Ning Lin[1]*

1 Department of Clinical Nutrition, The General Hospital of Western Theater Command, Chengdu, P. R. China, 2 School of Public Health, Southwest Medical University, Luzhou, P. R. China, 3 Wanzhou District Market Supervision Administration, Chongqing, P. R. China, 4 General Medicine, Chengdu Second People's Hospital, Chengdu, P. R. China

* helenmedic@yeah.net

**Data Availability Statement:** All raw data and blot/gel images have been uploaded as Supporting information.

## Abstract

### Background

*Penthorum chinense* Pursh (PCP), a medicinal and edible plant, has been reported to protect against liver damage by suppressing oxidative stress. Type 2 diabetes mellitus (T2DM) is associated with liver dysfunction and oxidative stress. In the present study, we aim to investigate the hypoglycemic effect of PCP on *db/db* mice and further explore the underlying mechanisms.

### Methods

Thirty-two *db/db* mice were randomized into four groups, including a diabetic model control group (MC) and three diabetic groups treated with low (LPCP, 300 mg/kg/d), medium (MPLP, 600 mg/kg/d), and high doses of PCP (HPCP, 1200 mg/kg/d), and the normal control group (NC) of eight *db/m* mice were included. Mice in the NC and MC groups received the ultrapure water. After four weeks of intervention, parameters of fasting blood glucose (FBG), insulin resistance (IR), blood lipid levels, hepatic oxidative stress, and enzymes related to hepatic glucose metabolism were compared in the groups.

### Results

PCP administration significantly reduced FBG and IR in diabetic *db/db* mice, and improved hepatic glucose metabolism by increasing glucose transporter 2 (GLUT2) and glucokinase (GCK) protein expression. Meanwhile, PCP supplementation ameliorated hepatic oxidative stress by decreasing malonaldehyde content and increasing the activities of superoxide dismutase and glutathione peroxidase in *db/db* mice. Furthermore, PCP treatment reduced obesity and food intake in *db/db* mice, and improved dyslipidemia demonstrated by increasing high-density lipoprotein cholesterol (HDL-C) while decreasing total cholesterol (TC), triglyceride (TG), and low-density lipoprotein cholesterol (HDL-C). All doses of PCP treatment decreased the values of LDL-C/HDL-C in a dose-response relationship.

**Funding:** This research was funded by grants from the Key Research and Development of the Science and Technology Plan Project of Luzhou City (2012-NYF-17), the Joint Research Project of the General Hospital of Western Theater Command (KY201900), and the Sichuan Cadre Healthcare Research Project (No. 2021-1303).

**Competing interests:** The authors have declared that no competing interests exist.

## Conclusion

PCP significantly alleviated hyperglycemia, hyperinsulinemia, hyperlipidemia, and obesity, inhibited hepatic oxidative stress, and enhanced hepatic glucose transport in T2DM mice. Based on the above findings, the hypoglycemic effect of PCP may be attributed to the activation of the GLUT2/GCK expression in the liver and the reduction of hepatic oxidative stress.

## Introduction

Diabetes mellitus (DM) is a chronic metabolic disease featured by hyperglycemia, which ranks in the top 10 causes of death worldwide. According to the report from the International Diabetes Federation, diabetes may increase from 463 million to 700 million by 2045 [1, 2]. Diabetes has become a severe health problem worldwide due to its high prevalence, disability, and mortality [3]. The characteristic symptoms of diabetes are polyuria, polydipsia, polyphagia, hyperhidrosis, and unexpected weight loss, in addition to hyperglycemia and abnormalities in serum lipids [4]. If not strictly controlled, chronic hyperglycemia can cause organ damage and serious complications such as angiocardiography, neuropathy, and nephropathy [5]. Type 2 diabetes mellitus (T2DM) is the most common form of diabetes, accounting for 90–95% of the diabetic population [6]. The etiology and pathogenesis of T2DM are not clear, but the distinctive pathophysiological feature is the reduced ability of insulin to regulate glucose metabolism manifested as insulin resistance (IR), and/or relatively reduced insulin secretion due to defective islet β-cell function.

Obesity/overweight, genetics, environment, and sedentary lifestyle may be factors associated with T2DM. A cross-sectional survey from China showed that the prevalence of diabetes among normal-weight, overweight, and obese people was 8.8%, 13.8%, and 20.1% respectively [7]. Moderate weight loss in overweight or obese people with T2DM can improve glycemic control and IR, as well as reduce the need for glucose-lowering medication [8]. Additionally, there is compelling evidence that oxidative stress contributes to more severe T2DM by reducing insulin secretion and function. Chronic hyperglycemia leads to the production of excess reactive oxygen species (ROS) in the body causing oxidative stress, which can result in disruption of insulin signaling in target tissues, as well as exacerbate IR and T2DM [9]. Several studies have shown that diabetic patients tend to exhibit higher malonaldehyde (MDA) content and lower superoxide dismutase (SOD) and glutathione peroxidase (GSH-Px) activity in serum and tissues. Furthermore, the levels of oxidative stress markers were reduced after treatment in patients with T2DM [10]. Therefore, the attenuation of oxidative stress is considered an effective strategy to improve IR and T2DM.

Most patients with T2DM require long-term oral medications or insulin injections to control blood glucose, however, their administration may be accompanied by adverse effects including gastrointestinal symptoms, hepatorenal functions, and hypoglycemia [11]. Thus, it is vital to search for natural, safe, and effective substitutes for these medicines.

Chinese edible herbal has the advantage of being more effective and having fewer side effects [12]. Penthorum chinense Pursh (PCP) is a traditional food and medicine widely distributed in East Asian countries. It has been demonstrated to possess potent antioxidant capabilities protecting against hepatic oxidative stress induced by alcohol and carbon tetrachloride [13, 14], but its effect on oxidative stress associated with T2DM has not been reported. In vitro studies found that ethyl acetate extract from PCP activates the Nrf2 antioxidant pathway by

binding to Keap1 protein, and significantly reduces the vascular inflammation induced by high glucose levels [15]. Meanwhile, a previous study had shown that PCP possessed a modest hypoglycemic effect by increasing insulin secretion in diabetic rats induced by streptozotocin (STZ) [16]. STZ is a broad-spectrum antibiotic with selective destruction of islet β-cells and is used to induce T1DM or T2DM models by varying doses of it. However, there is a large proportion of diabetic patients characterized by IR, whose insulin levels are equal to or even higher than normal. Consequently, the treatment strategy for this population is to increase insulin sensitivity rather than promote its secretion.

Genetic models, especially *db/db* mice, are regarded as the closest animal models to human T2DM with IR, which absorb excessive food chronically and spontaneously develop symptoms of obesity, IR, and hyperglycemia [17]. They are more like clinical patients in terms of diabetes production due to their lack of interference from artificially induced factors. To our knowledge, no study has investigated the hypoglycemic effects of PCP on *db/db* mice. This study aimed to observe the effects of PCP on hyperglycemia, IR, obesity, and oxidative stress in spontaneously T2DM mice, as well as explore the underlying mechanisms. The results of our study will provide more evidence for its application in the treatment of diabetes.

## Materials and methods

### Herbal materials and preparation

Dried PCP was obtained from Gulin Hongan Pharmaceutical Co., Ltd. (Luzhou, China). PCP samples were ground into powder and filtered through a 65-mesh sieve. Five hundred grams of the powder was immersed in 5000 mL ultra-pure water for 1h and exhaustively extracted at 100˚C for 3 rounds. The three filtrates were mixed, concentrated to 1 g/mL, and refrigerated at 4˚C.

### Animal experiments

All animal experiments were approved by the Ethics Committee of Southwest Medical University (No.201910-235). The 8-week-old male *db/db* and *db/m* mice were provided by Changzhou Cavens Co., Ltd. (Changzhou, China). All animals were fed in the SPF Animal Laboratory Center of Southwest Medical University at 12 h light/dark cycle, (23±2) ˚C, and (60±5) % relative humidity. All mice accessed food and water freely, whose padding was changed daily to keep them dry.

After seven days of acclimatization, the fasting blood glucose (FBG) was monitored by glucometer and glucose test strips (Roche Co., Germany). The *db/db* mice detected with FBG over 11.1 mmol/L at three executive mornings were defined as diabetic mice. Eight *db/m* mice were involved in the normal control (NC) group and forty diabetic *db/db* mice were randomly divided into four groups: model control (MC) group, low dose of PCP (LPCP) group, medium dose of PCP (MPCP) group and high dose of PCP (HPCP) group. The LPCP, MPCP, and HPCP groups were administered with a dose of 300, 600, and 1200 mg/kg/d, respectively. Allometric scaling to a person with a body weight of 60 kg, these doses correspond to 2-, 4-, and 8-gram PCP daily. 8 grams of PCP daily is the recommended upper intake level for humans. Only ultrapure water was given to the animals in the NC and MC. All treatments were administered by gavage once per day for four weeks.

Food consumption and body weight were measured weekly. All animals were given glucose (2g/kg) when an oral glucose tolerance test (OGTT) was conducted, and tail venous blood was collected at 0, 30th, 60th, and 120th minutes after oral gavage to measure blood glucose concentration.

## Biochemical analysis

After 4 weeks of intervention, mice were fasted the night before blood and tissue collection, then anesthetized with 1% sodium pentobarbital solution (1.0 mL/100g) according to Animal Veterinary Medical Association Guidelines for the Euthanasia of Animals, and executed. Glycated hemoglobin A1c (HbA1c) and insulin were detected by ELISA kits (Milian Co., Ltd., Shanghai, China). IR is indicated by the homeostasis model assessment of insulin resistance (HOMA-IR) with the calculation formula of FBG (mmol/L) × serum insulin (μU/mL)/22.5. The content of serum lipids and the hepatic antioxidant enzyme was tested by commercial kits (Jiancheng Co., Ltd., Nanjing, China).

## Hematoxylin and eosin staining in liver tissues

Biochemical Analyses. Liver tissue was fixed in a 4% paraformaldehyde solution and embedded in paraffin wax. The slices were sliced, dewaxed, hematoxylin, and eosin stained (H&E), and sealed in sequence. The vacuolar degeneration and steatosis of the hepatic tissue were observed under a light microscope.

## Western blotting

Liver tissue was lysed with RIPA buffer to produce lysate. Proteins were divided on 10% SDS-PAGE gels and moved to the PVDF membranes (Hybond, United States). The membranes were blocked with 5% skimmed milk for 120 minutes before being incubated with the primary antibody, and secondary antibody in sequence (Abcam, United Kingdom). Finally, the protein bands were tested by chemiluminescence kits (Affinity, United States) with an ECL chemiluminescence system (Tanon-5200Multi, Shanghai, China).

## Statistical analysis

All data were analyzed by SPSS 22.0 statistical software and the results are presented as the mean ± standard deviation (SD). One-way ANOVA analysis was performed to compare the mean differences among multiple groups. In pairwise comparison, the LSD test was used for those with homogeneous variances, and the Dunnett T3 test was used for those with uneven variances. A P-value of less than 0.05 was deemed statistically significant.

## Results

### PCP improved parameters related to hyperglycemia and IR in *db/db* mice

According to Fig 1A, the FBG in the NC was significantly lower than those in diabetic mice no matter before or after the intervention ($P<0.01$). In the second week, only the HPCP showed a significant decrease in FBG levels compared to MC ($P<0.05$). In the fourth week, the levels of FBG in all PCP-treatment groups significantly decreased to varying degrees compared to the MC ($P<0.05$). Compared with the MC, the HbA1c in diabetic mice of the LPCP, MPCP, and HPCP decreased in varying degrees, but with no significant difference (Fig 1B).

The results of OGTT were shown in Fig 1C. The levels of blood glucose among diabetic mice were always higher than that of the NC ($P<0.01$). After intragastric administration, the blood glucose of diabetic mice increased rapidly and reached the maximum values at the 30th minute. The blood glucose values in the HPCP were significantly lower than those in the MC at the 60th and 120th min ($P<0.01$, $P<0.05$). Additionally, high-dose PCP treatment significantly lowers the AUC of OGTT compared to that in the MC (Fig 1D, $P<0.01$).

The levels of insulin and HOMA-IR in each group were depicted in Fig 1E and 1F. Insulin and HOMA-IR of diabetic mice increased to different degrees when compared with the NC

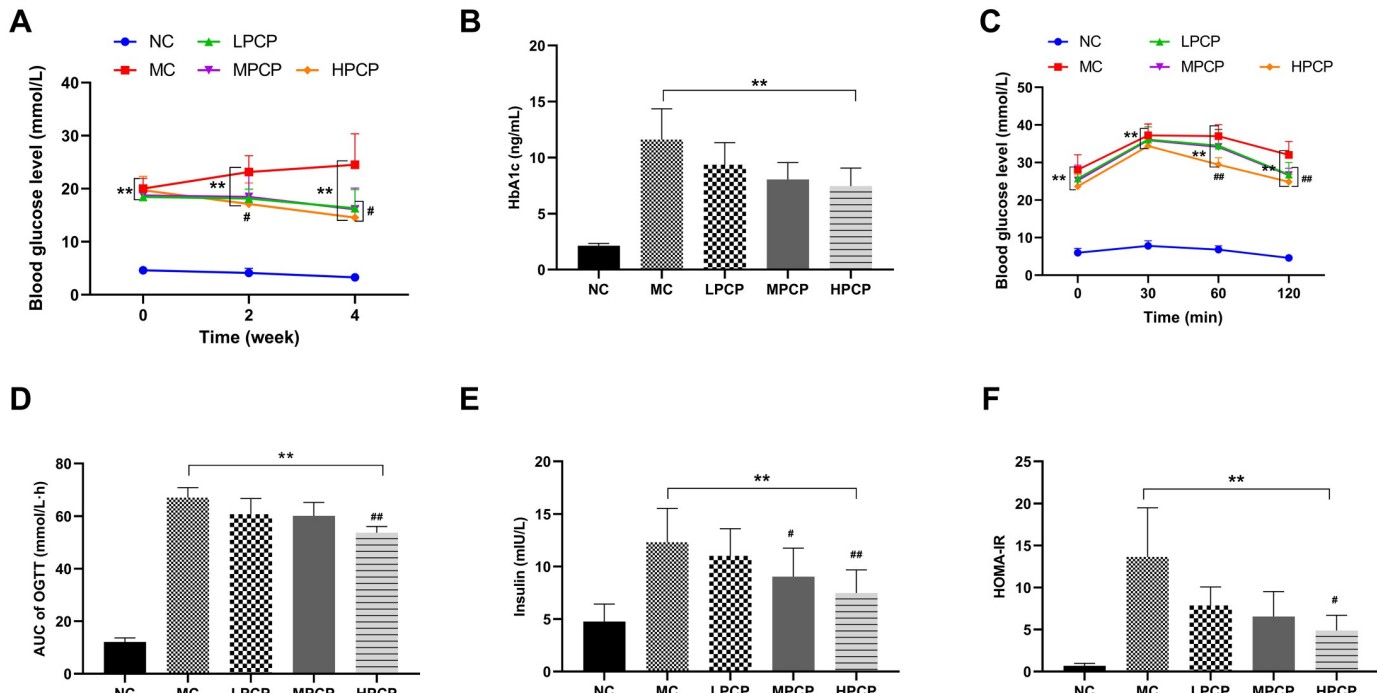

**Fig 1.** Effects of PCP on (A) FBG, (B) HbA1c, (C) OGTT, (D) AUC of OGTT, (E) Insulin, and (F) HOMA-IR in *db/db* mice. Data are expressed as mean ± SD; n = 8 for each experimental group. *$P<0.05$, **$P<0.01$ vs. normal control group. #$P<0.05$, ##$P<0.01$ vs. model control group.

($P<0.01$). Serum insulin was significantly alleviated after treatment with MPCP and HPCP in comparison to the MC group ($P<0.05$, $P<0.01$). Meanwhile, the HOMA-IR was significantly decreased by HPCP ($P<0.05$).

## PCP alleviated obesity in *db/db* mice

After PCP treatment, the food intake of diabetic mice was decreased in various degrees when compared to the MC (Fig 2A). The food intake of the HPCP was consistently significantly lower than the MC from the 1st to 4th week ($P<0.05$, $P<0.01$), which of the MPCP was significantly decreased compared with the MC from the 2nd to 4th week after the intervention ($P<0.05$, $P<0.01$). Initial body weights of diabetic mice were of no significant differences (Fig 2B). After 2 weeks of PCP treatment, the body weights of diabetic mice in PCP-treatment groups were significantly lower than the MC ($P<0.05$, $P<0.01$).

Both medium- and high-dose PCP treatments significantly decreased serum TG, while increased HDL-C ($P<0.01$, $P<0.05$) (Fig 2C–2F). Additionally, TC and LDL-C were significantly reduced in the HPCP ($P<0.05$, $P<0.01$). TG, TC, and LDL-C in LPCP were decreased by 8.84%, 11.06%, and 11.47%, as well as HDL-C was increased by 7.23%, but the changes failed to reach statistical significance after low-dose of PCP treatment ($P>0.05$). The results of lipid ratio parameters were shown in Fig 2G–2I. Those in the NC, including TC/HDL-C, TG/HDL-C, and LDL-C/HDL-C were significantly higher than those in the diabetic *db/db* mice ($P<0.01$). The values of TG/HDL-C in the MPCP and HPCP were significantly lower than those in the MC ($P<0.05$), while those of LPCP were not significantly different from those of the MC. However, the values of TG/HDL-C in the LPCP were significantly higher than those in the HPCP ($P<0.05$). All doses of PCP treatment decreased the values of LDL-C/HDL-C in a

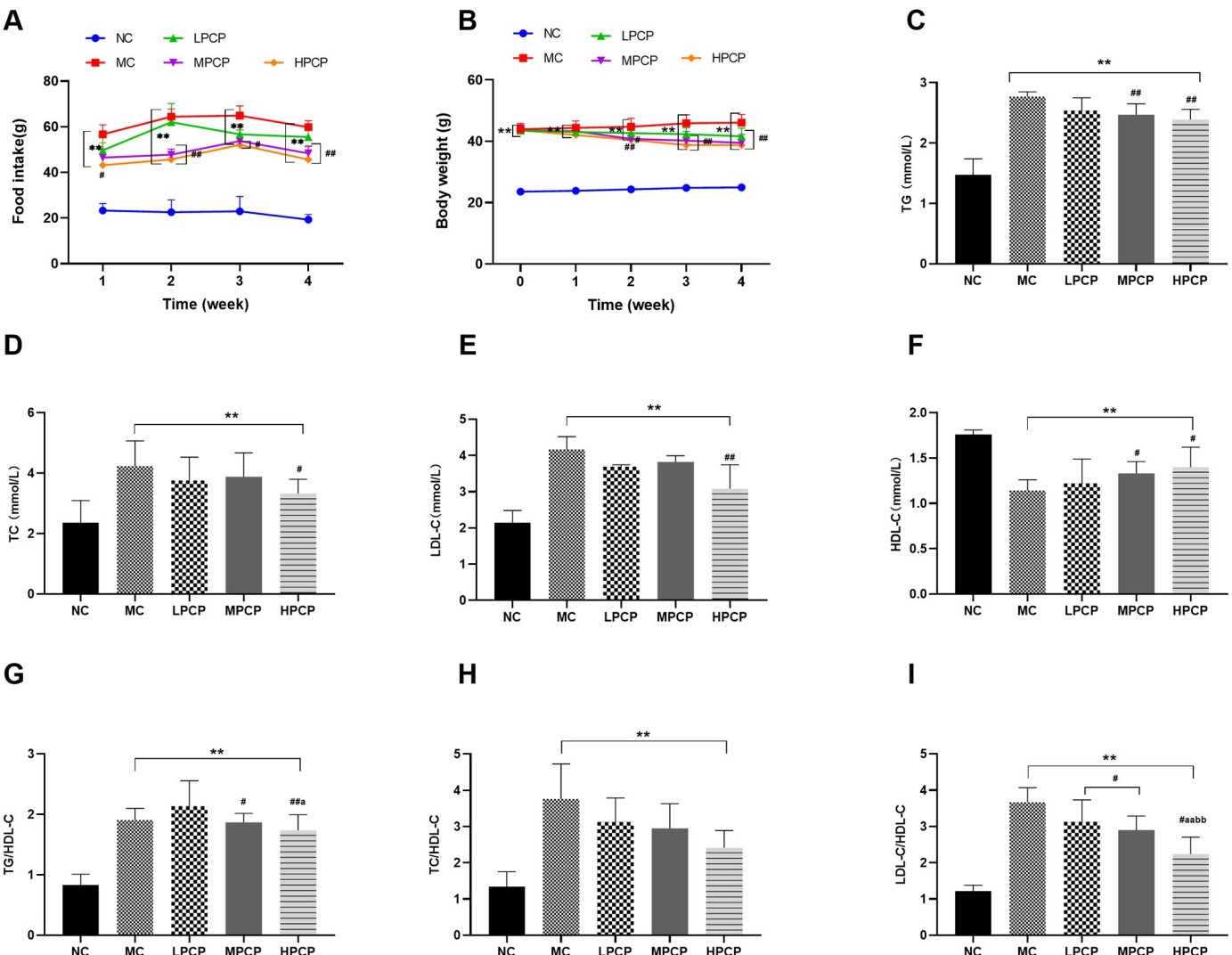

**Fig 2.** Effects of PCP on (A) food intake and (B) body weights, and serum lipid levels of (C) TG, (D) TC, (E) LDL-C, (F) HDL-C, (G) TG/HDL, (H) TC/HDL-C and (I) LDL-C/HDL-C in *db/db* mice. Data are expressed as mean ± SD; n = 8 for each experimental group. *$P<0.05$, **$P<0.01$ vs. normal control group. #$P<0.05$, ##$P<0.01$ vs. model control group. a$P<0.05$, aa$P<0.01$ vs. low dose of PCP. b$P<0.05$, bb$P<0.01$ vs. medium dose of PCP. c$P<0.05$, cc$P<0.01$ vs. high dose of PCP.

dose-response relationship ($P<0.05$), demonstrated by significantly lower values of LDL-C/ HDL-C in the HPCP than those in the LPCP and MPCP ($P<0.01$).

## PCP improved hepatic steatosis in *db/db* mice

H&E stains of the liver tissue were beneficial in evaluating hepatic cell fatty degeneration and injuries (Fig 3A and 3B). In the NC group, the hepatocytes with normal morphology were arranged neatly, and the structures of the hepatic lobules and sinus were intact and clear. Conversely, the hepatocyte cords in the MC group were disorganized and massive hepatocytes with hepatic steatosis or vacuolar degeneration were observed under the light microscope. After four weeks of treatment, the results showed that all doses of PCP had extremely ameliorated fatty or vacuolar degeneration in hepatic cells when compared with the MC group. In the LPCP, there was no steatosis in hepatocytes, although lots of vacuolar degeneration was found.

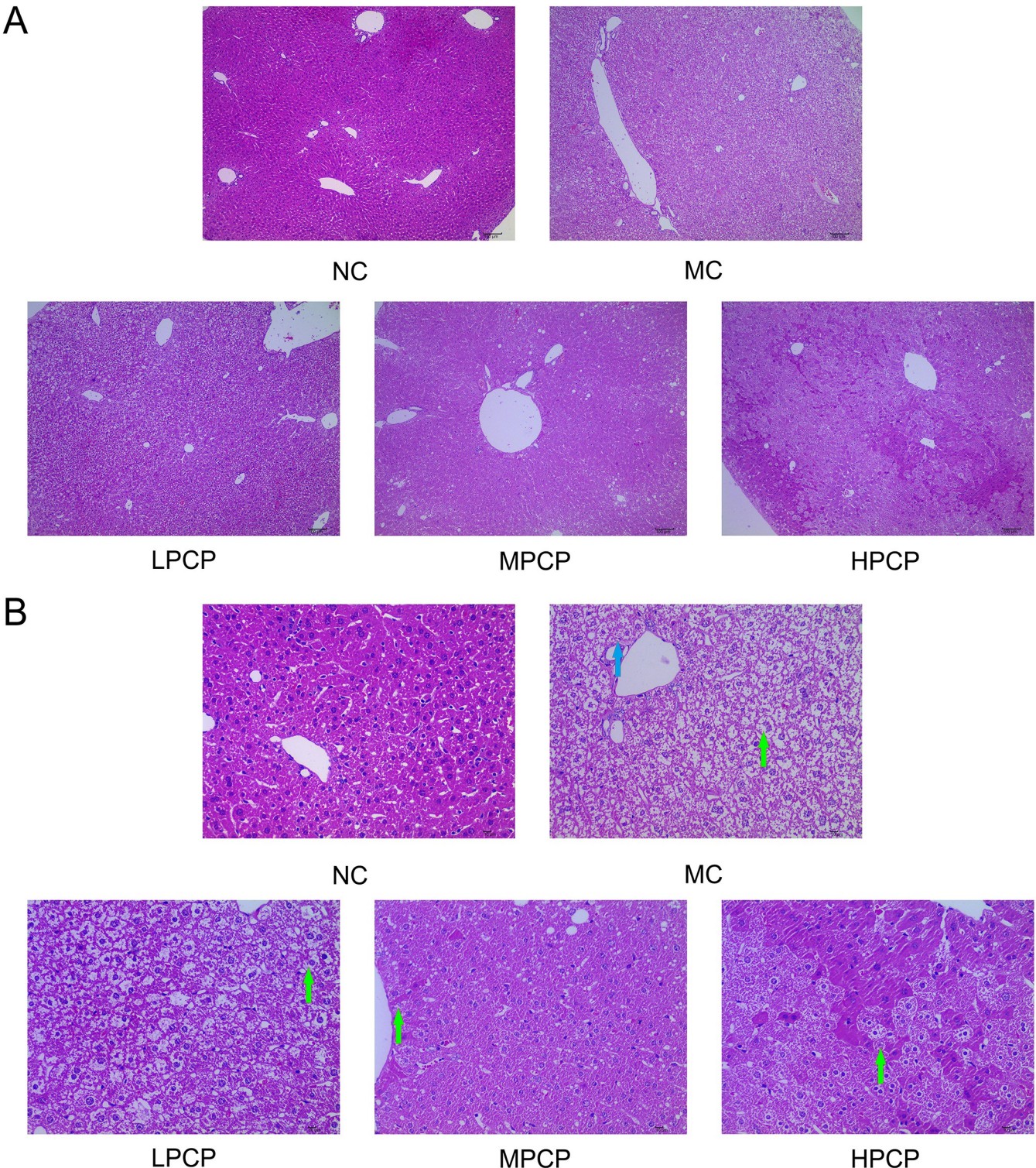

**Fig 3.** Representative images of H&E staining in liver tissues of each group (A) ×100, Scale bar, 100 μm; (B) ×400, Scale bar, 10 μm. ↑, steatosis; ↑, vacuolar degeneration.

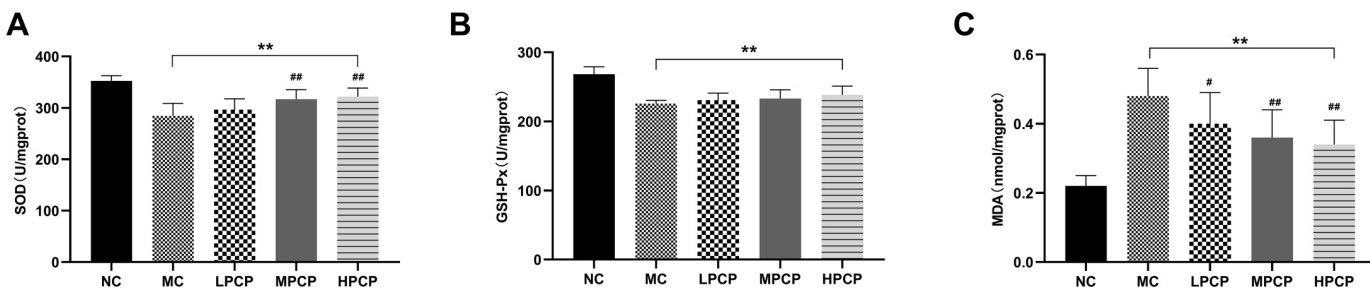

**Fig 4.** Effects of PCP on enzymatic antioxidant activities of (A) SOD and (B) GSH-Px, and the lipid peroxidation levels of (C) MDA in the liver of *db/db* mice. Data are expressed as mean ± SD; n = 8 for each experimental group. *$P<0.05$, **$P<0.01$ vs. normal control group. #$P<0.05$, ##$P<0.01$ vs. model control group.

In the MPCP, vacuolar degeneration in hepatocytes was significantly reduced. The hepatocytes in HPCP were arranged neatly and a little microvacuolar degeneration in hepatocytes was observed.

## PCP improved hepatic oxidative stress in *db/db* mice

Significant increase of activities of SOD in diabetic mice after medium- and high-dose of PCP treatment ($P<0.01$), together with a reduction of MDA in all PCP-treatment groups were presented by comparison with the MC ($P<0.05$, $P<0.01$) (Fig 4A–4C). Additionally, the LPCP, MPCP, and HPCP were found to increase GSH-Px activities of *db/db* mice by 2.31%, 3.31%, and 5.68%, respectively, when compared with the MC, but no significant differences were found among MC and PCP-treatment groups.

## PCP upregulated the protein expression of GLUT2 and GCK in *db/db* mice

As shown in Fig 5A and 5B, the protein expression of GLUT2 and GCK was significantly lower in diabetic *db/db* mice by comparison to the NC ($P < 0.01$, $P < 0.05$). However, treatment with PCP significantly increased the expression of GLUT2 and GCK compared to the MC ($P < 0.01$, $P < 0.05$). All doses of PCP increased GLUT2 protein expression in the liver, with HPCP being significantly more effective in upregulating GLUT2 expression than MPLP and LPCP ($P < 0.01$). Compared with the MC group, the relative expression of GCK protein increased by 22.55%, 28.51%, and 33.69% in the LPCP, MPCP, and HPCP groups, respectively ($P < 0.05$, $P < 0.01$), but it was not significant in PCP-treatment groups.

## Discussion

T2DM is characterized by abnormalities in glucose, lipid, and protein metabolism, as well as impairment in insulin secretion and/or IR [18]. In the present study, we demonstrated that PCP significantly reduced FBG, fasting insulin levels, and body weight in *db/db* mice, and ultimately ameliorated oral glucose tolerance, IR, dyslipidemia, and obesity. Meanwhile, we found that PCP inhibited hepatic oxidative stress and steatosis by increasing SOD and GSH-Px activities and reducing MDA content, with the high dose of PCP being significantly more effective. Mechanistically, the improvement of FBG and IR in *db/db* mice by PCP may be attributed to the activation of the GLUT2/GCK expression and reduction of hepatic oxidative stress.

Genetic models and a high-fat diet combined with low-dose STZ-induced models are probably the most used models of T2DM at present. *Db/db* mice, due to gene mutation of the leptin receptor, are insensitive to leptin and absorb excessive food chronically, spontaneously

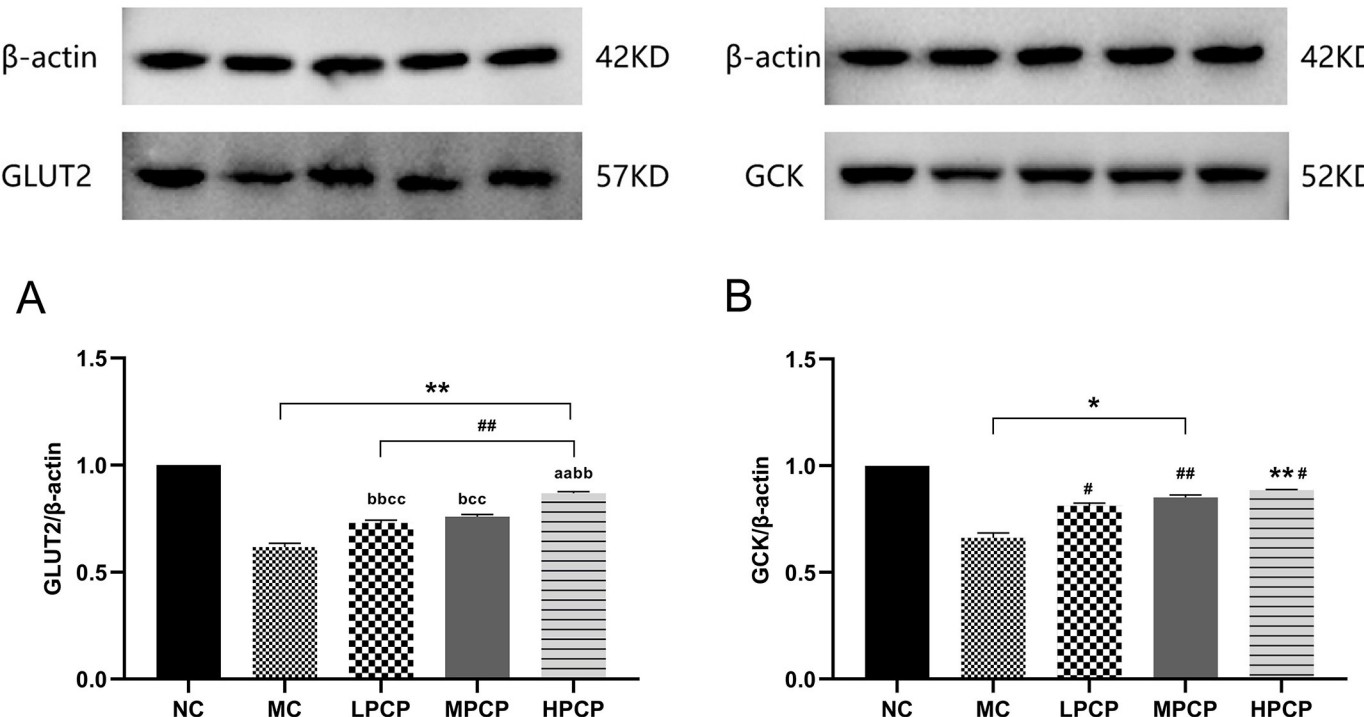

**Fig 5. Effect of PCP on glucose metabolism-related enzyme activities in the liver of *db/db* mice.** (A) The protein expression of GLUT2 in the liver expression of *db/db* mice; (B) The protein expression of GCK in liver tissue of *db/db* mice. $^*P<0.05$, $^{**}P<0.01$ vs. normal control group. $^\#P<0.05$, $^{\#\#}P<0.01$ vs. model control group. $^aP<0.05$, $^{aa}P<0.01$ vs. low dose of PCP. $^bP<0.05$, $^{bb}P<0.01$ vs. medium dose of PCP. $^cP<0.05$, $^{cc}P<0.01$ vs. high dose of PCP.

developing symptoms of obesity, IR, and hyperglycemia within the 8th week [19]. In comparison tests between the two models, *db/db* mice kept hyperglycemia in the whole experiment, whereas the blood glucose of STZ-induced models spontaneously declined after three weeks [20]. It was possible because a part of islet β-cells could repair themselves after STZ failure, which may affect the outcome of the experiment. Therefore, *db/db* mice were chosen for this study to remove the interference of human intervention on the results.

Insulin, secreted by islet β-cells located in the central part of the pancreas, is the only hormone with a hypoglycemic effect in the body [21]. The fasting serum insulin of *db/db* mice significantly declined after the intervention of PCP, and HOMA-IR was negatively correlated with the dose of PCP, which suggested that PCP significantly reduced IR in diabetic mice. Our findings were different from Huang's [16], which considered that PCP could enhance insulin secretion instead of reducing IR in diabetic rats. We further analyzed the reasons, which may be the different animal models chosen for the two studies. Their models were established by STZ, which is selectively toxic to the β-cells, to build diabetic models by disrupting the β-cells function and reducing insulin secretion [22]. In contrast, *db/db* mice are chronically overfed resulting in obesity, IR, and hyperglycemia, which occurs due to reduced insulin sensitivity rather than decreased insulin secretion [23].

IR is manifested by a decrease in insulin sensitivity that the normal amount of insulin secreted is insufficient to move glucose into target tissues including the liver, adipose, and muscle, which are commonly involved in the regulation of whole-body fuel metabolism [24, 25]. Previous studies have consistently concluded that PCP has a unique protective effect on the liver, which may be its target tissue [26]. The liver, a primary organ of glucose metabolism, is responsible for maintaining blood glucose homeostasis in the body. In hepatocytes, IR is

reflected by a decrease in glucose uptake and utilization, a reduction in hepatic glycogen synthesis, and an increase in hepatic glucose production and output, which subsequently leads to fasting hyperglycemia [27]. Non-alcoholic fatty liver disease (NAFLD) is the most common liver injury in diabetes and is characterized by excessive lipid deposition in hepatocytes. According to statistical data, the incidence of NAFLD in the general adult is 20–30%, but it occurs in obese and diabetic patients with an incidence of up to 70–80%. NAFLD is believed to promote insulin resistance and type 2 diabetes to form a vicious circle [28]. In this study, we observed that PCP significantly reduced liver steatosis or vacuolar degeneration in *db/db* mice. Similarly, Guo et al. [29] found that PCP exerted protective effects against hepatic steatosis and oxidative stress by enhancing the expression of AMP-activated protein kinase (AMPK). It has been demonstrated that the activation of AMPK in the liver can promote glucose metabolism and IR [30]. AMPK is a serine/threonine protein kinase, and the activated AMPK can increase glucose transporter protein expression and glucose uptake, and reduce IR to a greater degree [31]. Therefore, we further examined liver enzymes associated with glucose metabolism. GLUT2 and GCK are mainly secreted exclusively by hepatocytes and work together to maintain hepatic glucose metabolism homeostasis. GLUT-2 transports glucose in hepatic parenchymal cells, but the rate of glucose uptake is limited by the rate of glucose phosphorylation catalyzed by GCK [32]. Increasing the activity of GLUT2 and GCK inhibits hepatic glucose output and promotes glucose phosphorylation, hepatic glycogen synthesis, and aerobic metabolism of glucose in hepatocytes [33]. In the present study, PCP treatment significantly increased the expression of GLUT2 and GCK in *db/db* mice, with a high dose of PCP showing the best therapeutic effect and a dose-response relationship between the three groups. Based on the above findings, we proposed the hypothesis that PCP may decrease FBG and improve IR by upregulating the expression of GLUT2 and GCK to promote hepatic glucose uptake and utilization.

Hyperglycemia and IR are intimately linked to the overproduction of ROS and oxidative stress [34]. Sun et al. [15] found polyphenols from PCP show strong protective effects against high glucose-induced vascular inflammation by scavenging ROS and enhancing antioxidant activity, suggesting it to be a potential resource for anti-diabetes, which is consistent with our findings. Previous studies have shown that patients with T2DM produce large amounts of ROS during chronic hyperglycemia and that the liver is a major organ of ROS attack, disrupting its ability to scavenge free radicals and creating oxidative stress, thereby causing abnormalities in the insulin signaling pathway [35]. Therefore, the reduction of hyperglycemia-associated oxidative stress is considered to be an effective strategy to improve IR and T2DM [36]. Oxidative stress is usually assessed by measuring parameters including MDA, SOD, and GSH-Px. MDA, a product of lipid peroxidation, indirectly reflects the severity of free radical attack. SOD is responsible for catalyzing the breakdown of free radicals and reducing superoxide levels, which reflects the ability to scavenge free radicals [37]. The main role of GSH-Px is to protect the structural and functional integrity of cell membranes and to reduce the damage and attack of superoxide on the body [38]. In this study, the activities of SOD and GSH-Px were augmented and the levels of MDA were decremented in T2DM mice after PCP treatment, which is consistent with previous findings [39, 40]. These findings demonstrated that PCP can improve free radical scavenging ability by increasing SOD and GSH-Px activity and decreasing MDA content, thereby preventing liver damage under hyperglycemic conditions and regulating blood glucose and IR levels.

In addition to severe hyperglycemia, *db/db* mice exhibit severe symptoms including overeating and obesity [41]. There is strong evidence that obesity is among the most prominent hazard factors for IR and T2DM [42, 43]. Administration of PCP could potently reduce food consumption and alleviate body weight. No study discussed the effect of PCP on overeating

and obesity in *db/db* mice before our research. However, Li et al. found daily oral administration of PCP for 8 weeks decreased body weight, dyslipidemia, and insulin resistance in high-fat diet-induced non-alcoholic fatty liver disease (NAFLD) mice by regulating the gut microbiota and bile acid metabolism [44]. Furthermore, quercetin considered one of the key ingredients for PCP has been reported to restrain body weight gain and food intake in obese mice [45, 46]. We supposed that the anti-obesity effect of PCP might be moderately attributable to quercetin, but the hypothesis needs to be validated by further studies.

However, there are some limitations to our study. Firstly, PCP was only approved to be consumed in the form of tea bags in China. Therefore, *db/db* mice were selected to be intervened with an aqueous extract of PCP to simulate the way of human consumption. Secondly, in consideration of the fact that the markers of PCP are still unclear, we have not purified and identified its components. We will explore more deeply to determine the active components of PCP in alleviating hyperglycemia and IR. Finally, HbA1c in PCP treatment groups was lower than the MC, but no significant difference was observed. HbA1c is formed by continuous and irreversible glycosylation reaction and reflects the blood glucose levels in the previous two or three months [47]. We considered that the intervention time was too short to change HbA1c levels. Whether PCP treatment with a prolonged experimental period can improve the HbA1c of *db/db* mice needs further study.

## Conclusion

In summary, our results showed that PCP treatment decreased FBG and IR in T2DM mice. Moreover, it inhibited oxidative stress by promoting SOD and GSH-Px activities and reducing the MDA content. In addition, our results also showed that PCP regulated glucose transport and absorption by upregulating GLUT2 and GCK protein expression. PCP dose-dependently improved the parameters of oxidative stress and upregulated GLUT2 and GCK protein expression in comparison to the model group. Therefore, a high dose of PCP is more effective in alleviating IR, inducing hypoglycemic effects, and scavenging free radicals. The present study offers a new understanding of the health-promoting role of PCP and its potential significance in the therapy of T2DM.

## Supporting information

**S1 Raw data.**
(PDF)

**S1 Fig. Original image of western blot analysis.**
(PDF)

## Author Contributions

**Data curation:** Leyu Zheng, Xi Fan.

**Formal analysis:** Jilei Hu.

**Funding acquisition:** Huibo Xie, Ning Lin.

**Methodology:** Jilei Hu.

**Supervision:** Huibo Xie, Ning Lin.

**Writing – original draft:** Jilei Hu, Hongmei Lang.

**Writing – review & editing:** Huibo Xie, Ning Lin.

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
