## [Decision Letter · Decision Letter 0]

28 Jun 2024

PONE-D-24-18781Ameliorative Effects of Penthorum chinense Pursh on Insulin Resistance and Oxidative Stress in diabetic obesity db/db micePLOS ONE

Dear  Dr. Hu,

Thank you for submitting your manuscript to PLOS ONE. After careful consideration, we feel that it has merit but does not fully meet PLOS ONE’s publication criteria as it currently stands. Therefore, we invite you to submit a revised version of the manuscript that addresses the points raised during the review process. Please submit your revised manuscript by Aug 10 2024 11:59PM. If you will need more time than this to complete your revisions, please reply to this message or contact the journal office at plosone@plos.org. Please include the following items when submitting your revised manuscript:A rebuttal letter that responds to each point raised by the academic editor and reviewer(s). You should upload this letter as a separate file labeled 'Response to Reviewers'.A marked-up copy of your manuscript that highlights changes made to the original version. You should upload this as a separate file labeled 'Revised Manuscript with Track Changes'.An unmarked version of your revised paper without tracked changes. You should upload this as a separate file labeled 'Manuscript'.If applicable, we recommend that you deposit your laboratory protocols in protocols.io to enhance the reproducibility of your results. Protocols.io assigns your protocol its own identifier (DOI) so that it can be cited independently in the future. For instructions see: https://journals.plos.org/plosone/s/submission-guidelines#loc-laboratory-protocols. Additionally, PLOS ONE offers an option for publishing peer-reviewed Lab Protocol articles, which describe protocols hosted on protocols.io. Read more information on sharing protocols at https://plos.org/protocols?utm_medium=editorial-email&utm_source=authorletters&utm_campaign=protocols.

We look forward to receiving your revised manuscript.

Kind regards,

Mehran Rahimlou, PhD

Academic Editor

PLOS ONE

Journal Requirements:

2. To comply with PLOS ONE submissions requirements, in your Methods section, please provide additional information regarding the experiments involving animals and ensure you have included details on (1) methods of sacrifice, and (2) efforts to alleviate suffering.

4. In the online submission form, you indicated that all data generated or analyzed during this study are included in this published article. The raw data are available from the corresponding author on reasonable request.

Comments from PLOS Editorial Office: We note that one or more reviewers has recommended that you cite specific previously published works. As always, we recommend that you please review and evaluate the requested works to determine whether they are relevant and should be cited. It is not a requirement to cite these works. We appreciate your attention to this request.

Additional Editor Comments:

Dear Dr. Jilei Hu

Based on the comments received from the reviewers regarding this manuscript, we request you to revise your manuscript based on the comments raised by the reviewers.

Many thans

Reviewers' comments:

Reviewer's Responses to Questions

**Comments to the Author**

1. Is the manuscript technically sound, and do the data support the conclusions?

Reviewer #1: Yes

Reviewer #2: Yes

Reviewer #3: Yes

2. Has the statistical analysis been performed appropriately and rigorously? 

Reviewer #1: Yes

Reviewer #2: Yes

Reviewer #3: Yes

3. Have the authors made all data underlying the findings in their manuscript fully available?

Reviewer #1: Yes

Reviewer #2: Yes

Reviewer #3: Yes

4. Is the manuscript presented in an intelligible fashion and written in standard English?

Reviewer #1: Yes

Reviewer #2: Yes

Reviewer #3: Yes

5. Review Comments to the Author

Reviewer #1: 1.A few grammatical and typographical errors noted. For instance, ANIMALS EXPERIMENT, instead of ANIMAL EXPERIMENTS

2. De Ritis ratio and Ferritin could have been included

3. The authors are advised to compute lipid ratios, in view of their link with both diabetes mellitus and steatosis

4. Was the ELISA kit for HbA1c ,a high sensitivity and specificity one with minimal cross reactivity?? Why did the authors not use HPLC ???

Reviewer #2: 1. Language should be improved and spelling mistakes should be corrected.

2. Please abbreviate the text at first instance.

3. Give 2-3 lines about current status of diabetes at global level.

Brauer, Michael, Roth, Gregory A., Aravkin, Aleksandr Y., Zheng, Peng, Abate, Kalkidan Hassen, Abate, Yohannes Habtegiorgis, Abbafati, Cristiana, Kavita Munjal et al., 2024. Global burden and strength of evidence for 88 risk factors in 204 countries and 811 subnational locations, 1990–2021: a systematic analysis for the Global Burden of Disease Study 2021. The Lancet. 403, 2162-2203. https://doi.org/10.1016/S0140-6736 (24)00933-4.

Gauttam, V., Munjal, K., Mujwar, S., Sawale, J., Rohilla, M., Gupta, S. 2022. Comparative Study of Developed Formulation and Market Formulation for Antidiabetic Potential in Alloxan Induced Diabetic Wistar Rats. Journal of Young Pharmacists. 2022; 14(4), 387-393.

Reviewer #3: In this manuscript, the authors demonstrated that PCP significantly alleviated hyperglycemia, hyperinsulinemia, hyperlipidemia, and obesity, inhibited hepatic oxidative stress, and enhanced hepatic glucose transport in T2DM mice. Generally, the paper is well written and the results are of some scientific significance. I recommend this paper to be accepted after minor revision.

Minor points:

1. The inter-group significance shown in the picture is inconsistent with the description in the text. For example, in the result section ‘PCP improved parameters related to hyperglycemia and IR in db/db mice’ (Line 186), according to the figure, it should be ‘P<0.01’?

2. In Fig5-B, the significance may be wrong, especially HPCP Group. Please check through the manuscript to avoid these errors.”.

6. PLOS authors have the option to publish the peer review history of their article (what does this mean?). If published, this will include your full peer review and any attached files.

Reviewer #1: **Yes: **A.R.SRINIVASAN

Reviewer #2: **Yes: **Dr. Kavita Munjal

Reviewer #3: No

---

## [Author Response · Author response to Decision Letter 0]

3 Aug 2024

Dear Editor and Reviewers,

We sincerely thank you for your valuable feedback which helped to improve the quality of our manuscript entitled “Ameliorative Effects of Penthorum chinense Pursh on Insulin Resistance and Oxidative Stress in diabetic obesity db/db mice” (Manuscript ID: PONE-D-24-18781). We have considered your comments word by word and have made revisions based on your suggestions question by question. We have amended the format according to the Journal Requirements, and the details can be found in the flie of Point-by-point response to reviewers and editors.The reviewers’ comments are listed in blue italics and specific concerns have been numbered. Our response is given in normal font and changes/additions to the manuscript are marked in red. The main amendments in the paper and the responses to your comments are listed as follows:

For reviewer 1#

1. A few grammatical and typographical errors were noted. For instance, ANIMALS EXPERIMENT, instead of ANIMAL EXPERIMENTS.

Response: Thank you for your meticulousness. We felt sorry about this low-grade mistake and the amendment is in red.

2. De Ritis ratio and Ferritin could have been included.

Response: Thank you for your advice. Penthorum chinense Pursh (PCP) is widely utilized in China to treat a variety of liver diseases. Previous studies have demonstrated that the levels of aspartate aminotransferase (AST) and alanine aminotransferase (ALT) were ameliorated in a dose-dependent manner by PCP [1-4]. Therefore, we did not measure liver function-related indicators in this study and unfortunately, we were unable to calculate the De Ritis ratio either. As for ferritin, there is no published literature report on the effect of PCP on it. However, our team's ongoing but unpublished study suggests that total flavonoids of PCP were able to significantly attenuate liver injury of iron-overloaded rats by up-regulating the expression of divalent metal ion transporter protein 1 and transferrin receptor 1. Of course, if we can get new funds, we shall carry out more experiments to dig into the effect of PCP on ferritin.

3. The authors are advised to compute lipid ratios, because of their link with both diabetes mellitus and steatosis.

Response: Thank you for your advice. T2DM patients often present with characteristic plasma lipid and lipoprotein abnormalities, including low HDL-C, high LDL-C, and elevated TC and TG levels [5], which is consistent with our findings. Considering the reviewer’s suggestion, we further analyzed the lipid ratio parameters, including TC/HDL-C, TG/HDL-C, and LDL-C/HDL-C, that considered could predict the risk of T2DM complications by prior studies [6], especially for coronary heart disease. The results of lipid ratio parameters are shown in Fig 2G-I. Those in the NC, including TC/HDL-C, TG/HDL-C, and LDL-C/HDL-C were significantly higher than those in the db/db mice (P<0.01). The values of TG/HDL-C in the MPCP and HPCP were significantly lower than those in the MC (P<0.05), while those of LPCP were not significantly different from those of the MC. However, the values of TG/HDL-C in the LPCP were significantly higher than those in the HPCP (P<0.05). All doses of PCP treatment decreased the values of LDL-C/HDL-C in a dose-response relationship (P<0.05), demonstrated by significantly lower values of LDL-C/HDL-C in the HPCP than those in the LPCP and MPCP (P<0.01).

4. Was the ELISA kit for HbA1c, a high-sensitivity and specificity one with minimal cross-reactivity? Why did the authors not use HPLC?

Response: Thank you for your question. According to the literature, various methods developed for HbA1c examination include immunoassay, boronated affinity, enzymatic, capillary electrophoresis, and Ion-Exchange High-Performance Liquid Chromatography (IE-HPLC) [7], and each has its advantages and limitations. Currently, the two most used methods for HbA1c detection are IE-HPLC and immunoassay, and the results of them are in strong agreement in most cases. In addition to its advantages in detecting HbA1c, IE-HPLC methods can also detect other types of hemoglobin and the presence of hemoglobin variants, which are widely used in clinical practice. However, we did not choose IE-HPLC due to its high price. ELISA kits working on the principle of immunoassay, are one of the methods with high sensitivity, stable efficacy, and low price for the determination of HbA1c, which are widely used in animal experiments [8]. In summary, we finally chose the ELISA kits to measure the HbA1c levels of db/db mice.

For reviewer 2#:

1. Language should be improved and spelling mistakes should be corrected.

Response: Thank you for your suggestion. We have invited a native English speaker to help polish our manuscript, and we hope the revised manuscript will be acceptable to you.

2. Please abbreviate the text at the first instance.

Response: Thank you for your suggestion. We felt sorry about this low-grade mistake and now all full names with their abbreviations have been added when they first appear. The amendment is in red.

3. Give 2-3 lines about the status of diabetes at the global level.

Response: Thank you for providing valuable references. We have checked the literature carefully and added more description about the status of diabetes at the global level in the INTRODUCTION part of the revised manuscript (Lines 57-60).

For reviewer 3#:

1. The inter-group significance shown in the picture is inconsistent with the description in the text. For example, in the result section ‘PCP improved parameters related to hyperglycemia and IR in db/db mice’ (Line 186), according to the figure, it should be ‘P<0.01’.

Response: Thank you for your meticulousness. We felt sorry about this low-grade mistake and the amendment is in red.

2. In Fig5B, the significance may be wrong, especially HPCP Group. Please check through the manuscript to avoid these errors.

Response: Thank you for your meticulousness. We have checked and reanalyzed the raw data and unfortunately, the results and significance were real. As shown in Fig 5B, when compared to the MC, the relative expression of GCK protein increased in the LPCP, HPCP (P < 0.05), and MPCP (P < 0.01), respectively. However, there was no significant difference in multiple comparisons between the three PCP-treatment groups. We considered the following two possible reasons. Firstly, a smaller p-value does not mean that the difference is larger, only that more reason to think that there is a difference between the two groups. In this study, the relative expression of GCK protein in the MPCP and HPCP was significantly higher than the MC, but the p-value was smaller in the MPCP, which only means that there was more reason to think that the MPCP is different from the MC. Secondly, the above results may be due to the relatively small sample size of the experiment.

Finally, we would like to express our thanks again for your reviews and guidance on our manuscript. We sincerely hope that with the help of your knowledge, we can complete an excellent paper and we also hope our paper can be published in your journal.

Thank you and best regards.

Yours Sincerely,

Ning Lin

Clinical Nutrition, The General Hospital of Western Theater Command, Sichuan, Chengdu 610083, China.

Email: helenmedic@yeah.net

August 3, 2024

Reference

1. Du YC, Lai L, Zhang H, Zhong FR, Cheng HL, Qian BL, et al. Kaempferol from Penthorum chinense Pursh suppresses HMGB1/TLR4/NF-κB signaling and NLRP3 inflammasome activation in acetaminophen-induced hepatotoxicity. Food & function. 2020;11(9):7925-34.

2. Jiang Y, Zhong M, Zhan H, Tao X, Zhang Y, Mao J, et al. Integrated strategy of network pharmacology, molecular docking, HPLC-DAD and mice model for exploring active ingredients and pharmacological mechanisms of Penthorum chinense Pursh against alcoholic liver injury. Journal of ethnopharmacology. 2022;298:115589.

3. Wang S, Li W, Liu W, Yu L, Peng F, Qin J, et al. Total flavonoids extracted from Penthorum chinense Pursh mitigates CCl(4)-induced hepatic fibrosis in rats via inactivation of TLR4-MyD88-mediated NF-κB pathways and regulation of liver metabolism. Frontiers in pharmacology. 2023;14:1253013.

4. Zhang H, Cui X, Liu W, Xiang Z, Ye JF. Regulation of intestinal microflora and metabolites of Penthorum chinense Pursh on alcoholic liver disease. Frontiers in pharmacology. 2023;14:1331956.

5. Bahiru E, Hsiao R, Phillipson D, Watson KE. Mechanisms and Treatment of Dyslipidemia in Diabetes. Current cardiology reports. 2021;23(4):26.

6. Gu X, Yang X, Li Y, Cao J, Li J, Liu X, et al. Usefulness of Low-Density Lipoprotein Cholesterol and Non-High-Density Lipoprotein Cholesterol as Predictors of Cardiovascular Disease in Chinese. The American journal of cardiology. 2015;116(7):1063-70.

7. Rhea JM, Molinaro R. Pathology consultation on HbA(1c) methods and interferences. American journal of clinical pathology. 2014;141(1):5-16.

8. Huang D, Jiang Y, Chen W, Yao F, Huang G, Sun L. Evaluation of hypoglycemic effects of polyphenols and extracts from Penthorum chinense. Journal of ethnopharmacology. 2015;163:256-63.

---

## [Decision Letter · Decision Letter 1]

20 Sep 2024

Ameliorative Effects of Penthorum chinense Pursh on Insulin Resistance and Oxidative Stress in diabetic obesity db/db mice

PONE-D-24-18781R1

Dear Dr. Jilei Hu
,

We’re pleased to inform you that your manuscript has been judged scientifically suitable for publication and will be formally accepted for publication once it meets all outstanding technical requirements.

Kind regards,

Mehran Rahimlou, PhD

Academic Editor

PLOS ONE

Additional Editor Comments (optional):

According to the comments of two reviewers, the manuscript meets the necessary criteria for acceptance.

Reviewers' comments:

Reviewer's Responses to Questions

**Comments to the Author**

1. If the authors have adequately addressed your comments raised in a previous round of review and you feel that this manuscript is now acceptable for publication, you may indicate that here to bypass the “Comments to the Author” section, enter your conflict of interest statement in the “Confidential to Editor” section, and submit your "Accept" recommendation.

Reviewer #1: All comments have been addressed

Reviewer #3: All comments have been addressed

2. Is the manuscript technically sound, and do the data support the conclusions?

Reviewer #1: Yes

Reviewer #3: Yes

3. Has the statistical analysis been performed appropriately and rigorously? 

Reviewer #1: Yes

Reviewer #3: Yes

4. Have the authors made all data underlying the findings in their manuscript fully available?

Reviewer #1: Yes

Reviewer #3: Yes

5. Is the manuscript presented in an intelligible fashion and written in standard English?

Reviewer #1: Yes

Reviewer #3: Yes

6. Review Comments to the Author

Reviewer #1: Ameliorative Effects of Penthorum chinense Pursh on Insulin Resistance and Oxidative Stress in diabetic obesity db/db mice is a sound attempt. More light can be thrown by associating the markers of oxidative stress with the available and objective indicators of insulin sensitivity/insulin resistance, both dependent and independent of lipid status

Reviewer #3: The authors have made enough revisions and responses. The manuscript can be accepted in present form.

7. PLOS authors have the option to publish the peer review history of their article (what does this mean?). If published, this will include your full peer review and any attached files.

Reviewer #1: **Yes: **A.R.Srinivasan @ Vellore A.R.Srinivasan

Reviewer #3: No

---

## [Editor Report · Acceptance letter]

27 Sep 2024

PONE-D-24-18781R1 

PLOS ONE

Dear Dr. Hu, 

I'm pleased to inform you that your manuscript has been deemed suitable for publication in PLOS ONE. Congratulations! Your manuscript is now being handed over to our production team.

Kind regards, 

on behalf of

Dr. Mehran Rahimlou 

Academic Editor

PLOS ONE